# Cinnamaldehyde Decreases the Pathogenesis of *Aeromonas hydrophila* by Inhibiting Quorum Sensing and Biofilm Formation

**Shengping Li** [1,2], **Shun Zhou** [1], **Qiuhong Yang** [1], **Yongtao Liu** [1], **Yibin Yang** [1], **Ning Xu** [1], **Xiaohui Ai** [1,*] **and Jing Dong** [1,*]

1   Yangtze River Fisheries Research Institute, Chinese Academy of Fishery Sciences, Wuhan 430223, China
2   College of Fisheries and Life Science, Shanghai Ocean University, Shanghai 201306, China
*   Correspondence: aixh@yfi.ac.cn (X.A.); dongjing@yfi.ac.cn (J.D.); Tel.: +86-02781780010 (J.D.)

**Abstract:** Antibiotics were the main fishery drugs for treating *Aeromonas hydrophila* (*A. hydrophila*) infection, which would generate selective pressure and result in the appearance of antibiotic-resistant bacteria. The bacterial quorum sensing (QS) system provides a new alternative strategy against *A. hydrophila* infection. QS inhibitors can reduce bacterial virulence behaviors by disrupting QS, which has no effect on bacterial growth. Therefore, we studied the effect of cinnamaldehyde from a natural plant extract on the QS of *A. hydrophila* aiming to reduce its pathogenicity. The efficacy of cinnamaldehyde against *A. hydrophila* was evaluated from various aspects, including the effects on aerolysin, lipase, protease, swarming motility, biofilm formation, acyl-homoserine lactones (AHLs), and QS-related genes. Moreover, the therapeutic effect of cinnamaldehyde in vitro and in vivo was studied. The results showed that cinnamaldehyde could decrease the virulence phenotypes of *A. hydrophila* regulated by QS. Moreover, the transcriptions of related genes (*aerA*, *ahyR*, and *ahyI*) were downregulated following the addition of cinnamaldehyde. The in vitro and in vivo therapeutic assays show that cinnamaldehyde could reduce the aerolysin-mediated A549 cell injury and increase the survival rate of crucian carp infected with *A. hydrophila*. These results indicate that cinnamaldehyde would be a candidate QS inhibitor against *A. hydrophila* infection.

**Keywords:** *Aeromonas hydrophila*; quorum sensing; cinnamaldehyde; virulence; biofilm



## 1. Introduction

Aquatic products have become favorite foods for their rich, high-quality protein and the improvement of life quality, which accelerates the development of the aquaculture industry. Nevertheless, the fast growth of aquaculture brings a number of problems, particularly infectious diseases caused by bacteria. Antibiotics as the main drugs for fish health management and disease treatment were used in all stages of fish growth [1]. However, the overuse of antibiotics provided selective pressures to bacterial strains and resulted in the emergence of drug-resistant strains, which then decreased the drug efficacy or even resulted in treatment failure [2]. The invention of an anti-virulence strategy targeting bacterial quorum sensing (QS) provided a novel approach facing the incidence of antibiotic-resistant bacterial infections. The QS systems rely on signaling molecules secreted by bacteria that vary with bacterial density, which initiate the expression of related genes and further regulate their virulence behaviors and biofilm formation [3]. Therefore, the alternative drugs targeting QS can attenuate the pathogenicity and drug resistance of bacteria.

*A. hydrophila* is the main pathogen that threatens fish health and causes septicemia in fish [4]. *A. hydrophila* can secrete a variety of virulence factors, such as aerolysin, lipase, protease, and hemolysin, which determine the pathogenicity of the bacterium [5]. Moreover, *A. hydrophila* has a strong biofilm-forming ability, which can resist the effects of many

adverse conditions by attaching to surfaces, thereby promoting the drug resistance and pathogenicity of bacteria [6]. Studies have found that the virulence of *A. hydrophila* could be regulated by the QS [7]. Therefore, disrupting QS is an alternative approach to combating *A. hydrophila* infection.

Cinnamaldehyde (Figure 1A) is the main content of *Cinnamomum cassia* Presl, which exhibited a number of biological activities, such as antioxidant, antibacterial, antitumor, anti-inflammatory, and antidiabetic effects [8]. Studies have demonstrated that cinnamaldehyde performed against the growth of *A. hydrophila* by disrupting cell membranes and interfering with protein metabolism [9]. However, the function of cinnamaldehyde against the virulence of *A. hydrophila* has not been reported. Our study found that cinnamaldehyde could influence the pathogenicity of *A. hydrophila* by interfering with the QS.

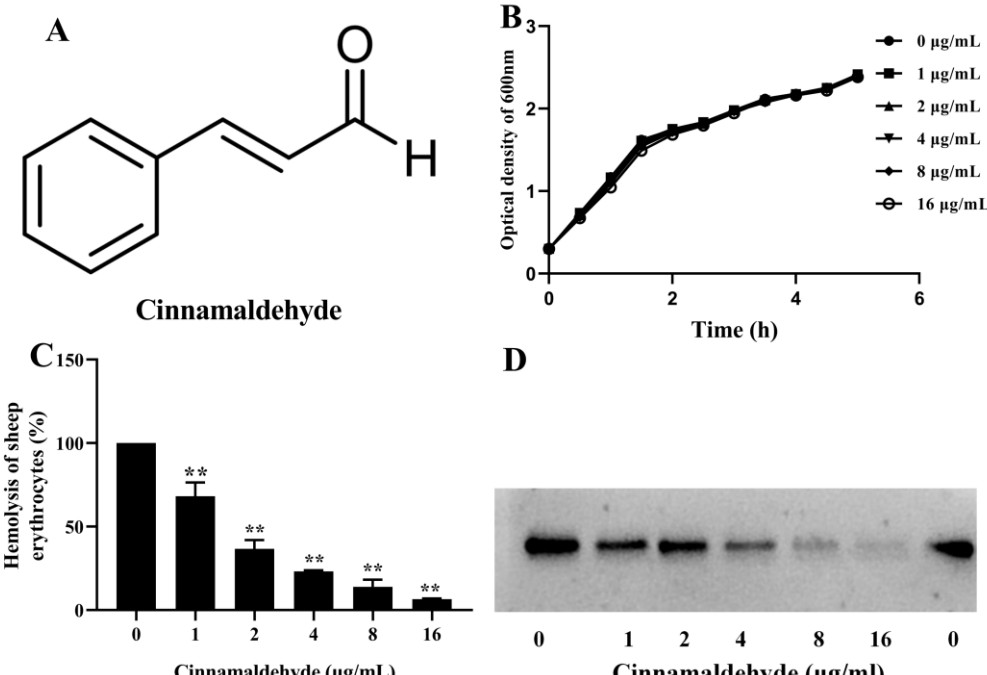

**Figure 1.** Inhibitory effects of cinnamaldehyde on hemolytic activity induced by aerolysin. (**A**) Chemical structure of cinnamaldehyde; (**B**) growth curves of *A. hydrophila* co-cultured with indicated concentrations of cinnamaldehyde; (**C**) effect of cinnamaldehyde on the hemolytic activity of bacterial supernatants; sheep erythrocytes treated with the bacterial supernatant of the drug-free group were used as the positive control; (**D**) detection of aerolysin production in bacterial supernatant plus cinnamaldehyde. The hemolysis assays in Figure 1C consist of data from three independent experiments. The results shown are the mean ± SD of the three experimental data compared to the drug-free group, **, $p < 0.01$.

## 2. Materials and Methods

### 2.1. Microorganisms and Reagents

*A. hydrophila* XS-91-4-1 and *Chromobacterium violaceum* CV026 were stored in our lab. Cinnamaldehyde (purity ≥ 95%) was purchased from Shanghai Aladdin Biochemical Technology Co., Ltd. Enrofloxacin was a commercial product purchased from the National Institutes for Food and Drug Control (Beijing, China). Cinnamaldehyde and enrofloxacin were dissolved in DMSO to obtain stock solutions of 40,960 µg/mL for in vitro assays and dissolved in 10% Tween 80 to obtain cinnamaldehyde emulsion for in vivo study.

### 2.2. Determination of Minimum Inhibitory Concentrations (MICs)

The MICs were determined according to a previously published protocol [10]. Briefly, cinnamaldehyde was diluted from 512 µg/mL to 2 µg/mL with MHB medium by 2-fold

dilution in a 96-well plate, while enrofloxacin was diluted from 32 μg/mL to 0.125 μg/mL. Then, the bacterial cells at a density of $5 \times 10^5$ CFU/mL were added to each well of the plate and cultured at 28 °C overnight. The wells without visible bacterial growth were defined as MICs.

### 2.3. Growth Curves Assay

Bacterial suspension cultured in LB broth was equally divided into 5 flasks when the optical density (OD) at 600 nm reached 0.3. Then, cinnamaldehyde was added into each flask to cause the final concentrations of cinnamaldehyde in the mixtures to reach 0, 1, 2, 4, 8, and 16 μg/mL, respectively. The mixtures were incubated at 28 °C for 5 h, and samples were taken every 30 min to read the $OD_{600nm}$ values.

### 2.4. Hemolysis

The hemolysis was measured using bacterial supernatants, as previously described [11]. Briefly, bacterial cultures at $OD_{600nm}$ of 0.3 were added to indicated concentrations of cinnamaldehyde (0, 1, 2, 4, 8, 16 μg/mL) and then co-cultured to $OD_{600nm}$ of 1.5. The bacterial suspensions were then centrifuged and then the supernatants were activated with the addition of trypsin, 100 μL of activated supernatant, 25 μL of sheep erythrocytes, and 875 μL of hemolysis buffer to obtain a 1 mL reaction system. The systems were then incubated at 37 °C for 20 min. The hemolytic activity of each drug group was read at $OD_{543nm}$ with a spectrophotometer.

### 2.5. Immuno-Blot

The concentrations of total proteins of bacterial supernatants described above were determined using the bicinchoninic acid method [12]. Then, the supernatants were sampled with Laemmli buffer and boiled for 10 min, then loaded for protein electrophoresis. A semi-dry transfer cell was used to transfer the proteins to the PVDF membrane. Then, the PVDF membrane was blocked with milk for 2 h, followed by incubation with the primary anti-aerolysin antibody and HRP-conjugated secondary goat anti-rabbit antiserum for 1 h, respectively. The levels of aerolysin in the supernatants were detected by an ECL detection kit.

### 2.6. Lipase Assay

Lipase assay was determined using the method reported by Ramanathan et al. [13]. In brief, the substrate mixture was composed of 0.3% (*w/v*) p-nitrophenyl palmitate in isopropanol and 50 mM $Na_2HPO_4$ buffer in an appropriate ratio (1:9). A total of 100 μL bacterial supernatants after different treatments were added to 900 μL of substrate mixture and incubated for 1 h. Then, the reaction was terminated by the addition of 1 M sodium carbonate buffer at the same volume of the system. After centrifugation, the values of $OD_{410nm}$ were determined in each reaction system.

### 2.7. Protease Activity Assay

According to a previous study, azocasein was used as the substrate [14]. Briefly, the supernatants of *A. hydrophila* co-cultured with cinnamaldehyde at concentrations ranging from 1 to 16 μg/mL were incubated with 20 mg/mL azocasein for 1 h, and 100 g/L trichloroacetic acids were added to the mixture to precipitate the protein. Then, an equal volume of NaOH (1 mol/L) was added, and the values were read at the OD of 440 nm.

### 2.8. Swarming Motility Assay

Sterilized swarming agar consists of 1% glucose, 0.5% agar, 0.5% peptone, and 0.2% yeast extract [15]. A total of 15 mL of swarming agar with different concentrations of cinnamaldehyde (1, 2, 4, 8, and 16 μg/mL) were added to the plates, and then 2 μL of bacterial suspension ($OD_{600nm}$ = 1.0) was added in the middle of the agar. Swarming agar

plus DMSO was set as the control group. All plates were incubated for 24 h at 28 °C and the diameters of swarming were determined.

### 2.9. Biofilm Formation

Cinnamaldehyde was 2-fold diluted at volumes of 100 μL in a 96-well plate, and then bacterial cultures at volumes of 100 μL were added to each well after a dilution of 1:10. After incubation at 28 °C for 24 h, the cultures in 96-well plates were washed twice with PBS to remove unattached cells. The plate was air-dried and 0.5% crystal violet was added to stain bacterial cells. After washing, 95% ethanol was added into each well to release crystal violet in bacterial cells, and the absorption values were further determined at $OD_{570nm}$.

For microscopic analysis, cinnamaldehyde and bacterial cultures were added to the 24-well plate containing glass slides and then incubated for 24 h. Bacterial cells attached in glass slides were stained with crystal violet after washing and images were pictured under a microscope.

### 2.10. qPCR Assay

Bacterial cells collected in hemolysis assay were used for qPCR assay. Briefly, a commercial RNA isolation kit was used to obtain the total RNA of the samples, then DNA remaining in the total RNA was removed. After reverse transcription, qPCR reactions were carried out to determine the expression levels of target genes. Primer pairs used in the study were listed in Table 1. Consequently, 16s rRNA was defined as the internal standard. Ct values were obtained and the expression levels were calculated by $2^{-\Delta\Delta Ct}$.

**Table 1.** Primer pairs used in qPCR assay.

| Primer | Sequence | PCR Amplicon (bp) | Accession No. |
|---|---|---|---|
| *aerA*-F | TCTACCACCACCTCCCTGTC | 218 | NC008570.1 |
| *aerA*-R | GACGAAGGTGTGGTTCCAGT | | |
| *ahyI*-F | GTCAGCTCCCACACGTCGTT | 202 | CP000462.1 |
| *ahyI*-R | GGGATGTGGAATCCCACCGT | | |
| *ahyR*-F | TTTACGGGTGACCTGATTGAG | 206 | CP000462.1 |
| *ahyR*-R | CCTGGATGTCCAACTACATCTT | | |
| 16S rRNA-F | TAATACCGCATACGCCCTAC | 164 | NR074841.1 |
| 16S rRNA-R | ACCGTGTCTCAGTTCCAGTG | | |

### 2.11. AHLs Production Assay

The effect of cinnamaldehyde on the AHLs production of *A. hydrophila* was determined by the previous method [16]. Briefly, *A. hydrophila* and *C. violaceum* CV026 cultured overnight were streaked in parallel at equal intervals on agar plates plus indicated concentrations of cinnamaldehyde at 28 °C for 24 h. To quantify violacein production, *A. hydrophila* was co-cultured with cinnamaldehyde for 24 h, then bacterial supernatants were collected and sterilized. An overnight *C. violaceum* CV026 culture was sub-inoculated into the sterilized supernatants and further cultured for 24 h. Then, bacterial cells of *C. violaceum* CV026 were collected by centrifugation and DMSO was added to release violacein pigment. The production of violacein was determined by a microplate reader at 585 nm.

### 2.12. Cell Viability Assays

A549 cells cultured in DMEM plus 10% fetal bovine serum and 5% $CO_2$ were used to determine the protective effect of cinnamaldehyde against aerolysin-induced cell injury. Cells were seeded into a 96-well plate after being digested by trypsin. Sterile bacterial supernatants after treatment with cinnamaldehyde were co-cultured with the cells for 1.5 h. Then, the cells were acquired for live/dead cell staining, and cell-free supernatants for LDH release assays, respectively. The percent of LDH release was calculated by determining

the values of $OD_{490nm}$. Cell injury was evaluated by photographing cells treated with live/dead regents.

### 2.13. Animal Study

Animal studies were performed under the guidance of the Animal Welfare and Research Ethics Committee at the Yangtze River Fisheries Research Institute (Permission No. YFI-2022DJ-011, 10 June 2022). All the experimental protocols were approved and supervised by the animal care committee. A total of 60 crucian carps weighing $200 \pm 10$ g were maintained in 100 L glass tanks for 7 days and then were separated into 3 groups. For the experimental conditions, the dissolved oxygen in culture water was more than 5.0 mg/L, the test water temperature was $28 \pm 2$ °C, and the pH was 7.5~8.0. *A. hydrophila* XS-91-4-1 was cultured to the mid-log phase and diluted to a concentration of $1.5 \times 10^8$ CFU/mL with sterile PBS. The fish infection model was established with the injection of the bacterial suspension at a volume of 100 μL intraperitoneally, and 100 μL of the sterile PBS for the negative control group. The cinnamaldehyde treatment group was given 25 mg/kg cinnamaldehyde by a gavage needle, while the positive and negative control group were administered with 10% Tween 80. The course was maintained for 3 days at 12-h intervals. The mortality in each group was recorded every day for 8 days. Three independent tests were carried out in this experiment.

### 2.14. Statistical Analysis

Statistical analysis was performed using GraphPad Prism 8.0 software. Data on survival rate were analyzed by Kaplan–Meier estimates and log-rank test, while other data were firstly analyzed by one sample K-S and explore tests, and then by Student's *t*-test to determine the statistical significance. $p < 0.05$ indicates statistical significance.

## 3. Results

### 3.1. Effect of Cinnamaldehyde on A. hydrophila Growth

The MICs were 128 μg/mL and 4 μg/mL for cinnamaldehyde and enrofloxacin, respectively. The results indicated that cinnamaldehyde had little anti-*A. hydrophila* activity. Moreover, the growth curves assay was performed to evaluate the influence of cinnamaldehyde on bacterial growth in 5 h. As shown in Figure 1B, cinnamaldehyde had no role in bacterial growth when the concentrations reached 1 to 16 μg/mL. Furthermore, the addition of DMSO in the drug-free group showed that DMSO could not inhibit bacterial growth. Taken together, the results demonstrated that *A. hydrophila* co-cultured with cinnamaldehyde under our experimental conditions could not provide selective pressure to the bacterium.

### 3.2. Cinnamaldehyde Inhibited the Hemolysis of A. hydrophila

The hemolysis results showed that cinnamaldehyde could reduce the hemolytic activity of bacterial supernatants obtained from *A. hydrophila* plus cinnamaldehyde dose-dependently (Figure 1C). The hemolytic activity declined to $68.14 \pm 8.25$, $36.64 \pm 5.31$, $23.21 \pm 0.60$, $13.94 \pm 4.30$, and $6.50 \pm 0.42\%$ when co-cultured with cinnamaldehyde at concentrations of 1, 2, 4, 8, and 16 μg/mL compared with the cinnamaldehyde-free group. Hemolysis was remarkably inhibited by the addition of cinnamaldehyde at concentrations higher than 1 μg/mL. Moreover, the immune-blot assay was conducted to determine the relationship between aerolysin production and hemolysis. As expected, aerolysin production decreased with increasing drug concentrations (Figure 1D). Therefore, cinnamaldehyde could decrease the hemolysis of *A. hydrophila* supernatants by reducing the production of aerolysin.

### 3.3. Inhibitory Effect on Lipase Production

Cinnamaldehyde could dose-dependently reduce lipase production of *A. hydrophila* when the concentrations of cinnamaldehyde range from 4 to 16 μg/mL (Figure 2A). Com-

pared with the cinnamaldehyde-free group, lipase production reduced to 73.49 ± 5.91% when co-cultured with 16 μg/mL cinnamaldehyde.

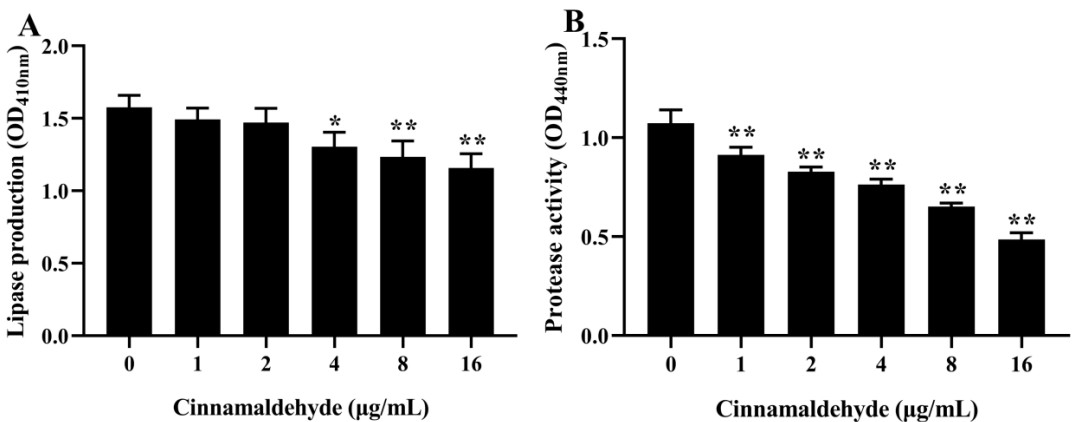

**Figure 2.** Inhibitory effect of cinnamaldehyde on two extracellular enzymes of *A. hydrophila*. (**A**) Lipase production of *A. hydrophila* plus cinnamaldehyde; (**B**) the effect of cinnamaldehyde on the protease activity of *A. hydrophila*. The data in both assays are mean ± SD of three independent experiments. *, $0.01 < p < 0.05$ and **, $p < 0.01$ when compared with drug-free group.

### 3.4. Inhibitory Effect on Protease Activity

As shown in Figure 2B, cinnamaldehyde could dose-dependently inhibit the protease activity of *A. hydrophila*. Compared with the drug-free group, the protease activity decreased to 85.24 ± 5.59, 7.22 ± 2.47, 71.14 ± 2.19, 60.87 ± 2.16, and 45.28 ± 1.50%, plus cinnamaldehyde at concentrations ranging from 1 to 16 μg/mL.

### 3.5. Inhibitory Effect of Cinnamaldehyde on Swarming Motility

As shown in Figure 3A, the swarming zone of *A. hydrophila* was visible on drug-free agar, while the swarming range became narrow following the addition of cinnamaldehyde at indicated concentrations (Figure 3B–F). Moreover, the swarming diameter was determined. As shown in Figure 3G, the swarming diameter decreased to 10.67 ± 1.25, 9.33 ± 0.47, 8.33 ± 0.47, 7.00 ± 1.41, and 4.67 ± 0.47 mm, and 15.33 ± 1.89 mm for the drug-free group. The swarming motility was significantly reduced when co-cultured with cinnamaldehyde at concentrations ranging from 1 to 16 μg/mL.

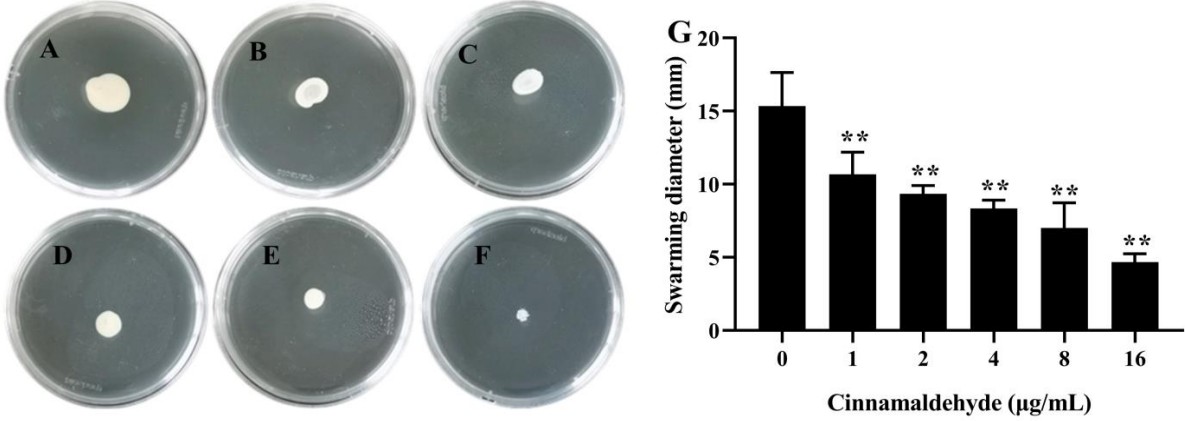

**Figure 3.** Effects of cinnamaldehyde on swarming motility. (**A**) DMSO control group; (**B**) 1 μg/mL; (**C**) 2 μg/mL; (**D**) 4 μg/mL; (**E**) 8 μg/mL; (**F**) 16 μg/mL; (**G**) swarming diameter. All data are presented as mean ± SD in the swarming diameter of three independent experiments. **, $p < 0.01$ when compared to the DMSO group.

### 3.6. Inhibition of Biofilm Formation

As shown in Figure 4A, cinnamaldehyde could reduce *A. hydrophila* biofilm formation in a dose-dependent manner. The inhibition rates of biofilm formation ranged from $17.54 \pm 11.62\%$ to $70.26 \pm 8.22\%$, as the drug concentration was increased from 1 to 16 µg/mL. Moreover, the biofilm on glass slides was observed by microscopy. As shown in Figure 4B, a large number of bacterial cells stained purple could be observed on a drug-free slide, while a few cells were presented in the 16 µg/mL drug group (Figure 4C).

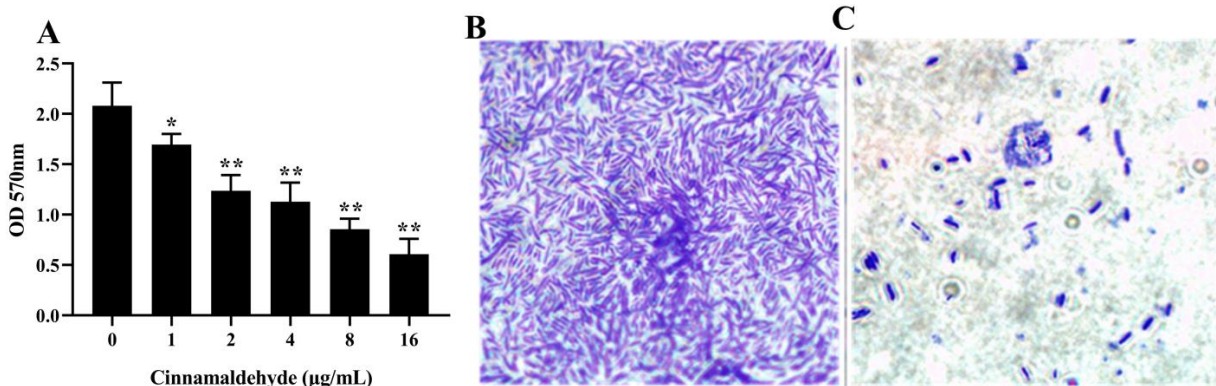

**Figure 4.** Effect of cinnamaldehyde on biofilm formation of *A. hydrophila*. (**A**) Determination of the cinnamaldehyde effect on bacterial biofilms, all data are the mean $\pm$ SD of three independent experiments. *, $0.01 < p < 0.05$ and **, $p < 0.01$ when compared with drug-free group; (**B**) cinnamaldehyde-free group; (**C**) 16 µg/mL.

### 3.7. Cinnamaldehyde Reduced the Transcription of Related Genes

The above results showed that cinnamaldehyde could reduce the production of virulence factors and biofilm formation, which are regulated by the QS system of *A. hydrophila*. Therefore, we determined the effect of cinnamaldehyde on aerolysin coding genes (*aerA*) and QS-related genes (*ahyR, ahyI*). As shown in Figure 5, the transcription of *aerA* gene was reduced to 5.58-fold under the 16 µg/mL drug compared with the drug-free group, and 13.48-fold and 8.35-fold for *ahyI* and *ahyR*, respectively. Moreover, statistical significance was observed in the lowest tested concentration of 1 µg/mL.

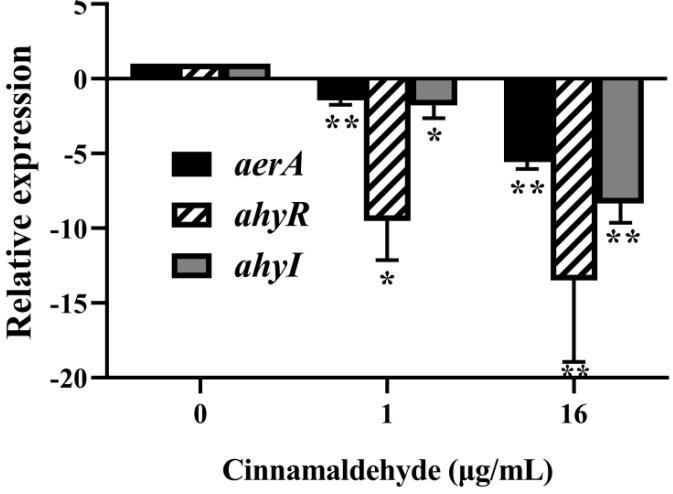

**Figure 5.** The effects on related genes of *A. hydrophila* plus cinnamaldehyde. Three independent qPCR assays were performed for *aerA*, *ahyI*, and *ahyR* genes, and the data are mean $\pm$ SD. *, $0.01 < p < 0.05$ and **, $p < 0.01$ when compared to drug-free group.

### 3.8. Cinnamaldehyde Reduced AHLs Production

qPCR results showed that cinnamaldehyde could downregulate the transcript of related genes, while AHLs are the main signal molecules in the QS system regulating the expression of related genes of *A. hydrophila*. The results indicated that cinnamaldehyde might inhibit AHLs production. Thus, bioreporter *C. violaceum* CV026 was used to investigate the production of AHLs of *A. hydrophila* after a co-incubation with indicated concentrations of cinnamaldehyde. As shown in Figure 6A, *C. violaceum* CV026 produced a visible purple color in the drug-free agar, while a little purple color was produced on 16 μg/mL of the drug agar (Figure 6F). Moreover, the violacein production of *C. violaceum* CV026 was quantitatively determined by releasing it in DMSO. As shown in Figure 6G, cinnamaldehyde could dose-dependently reduce violacein production in *C. violaceum* CV026, and statistical significance was observed in the 2 μg/mL drug group.

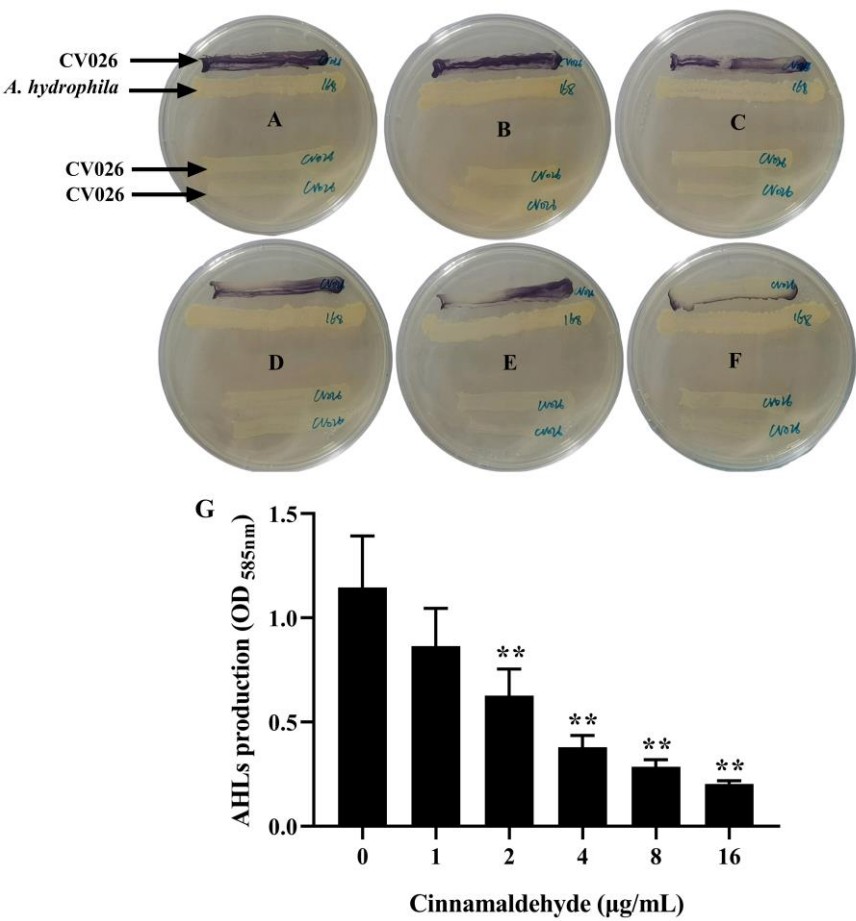

**Figure 6.** Cinnamaldehyde reduces AHLs production by *A. hydrophila*. (**A**) Cinnamaldehyde-free group; (**B**) 1 μg/mL; (**C**) 2 μg/mL; (**D**) 4 μg/mL; (**E**) 8 μg/mL; (**F**) 16 μg/mL; (**G**) violacein production of *C. violaceum* CV026 in response to AHLs at indicated cinnamaldehyde concentrations, all data are the mean ± SD of three independent experiments. **, $p < 0.01$ when compared to the drug-free group.

### 3.9. Cell Viability Results

As shown in Figure 7A, untreated cells were stained green indicating live cells, while cells after treatment with a drug-free supernatant indicating dead cells were stained red (Figure 7B). The ratio of dead cells in the visual field was obviously decreased compared with cells in the cinnamaldehyde-free group (Figure 7C). Moreover, LDH release results showed that LDH release reduced with increasing concentrations of cinnamaldehyde (Figure 7D). When co-cultured with 16 μg/mL drug-treated supernatant, LDH release decreased to 38.12 ± 2.12%, and to 67.50 ± 3.77% for the drug-free group.

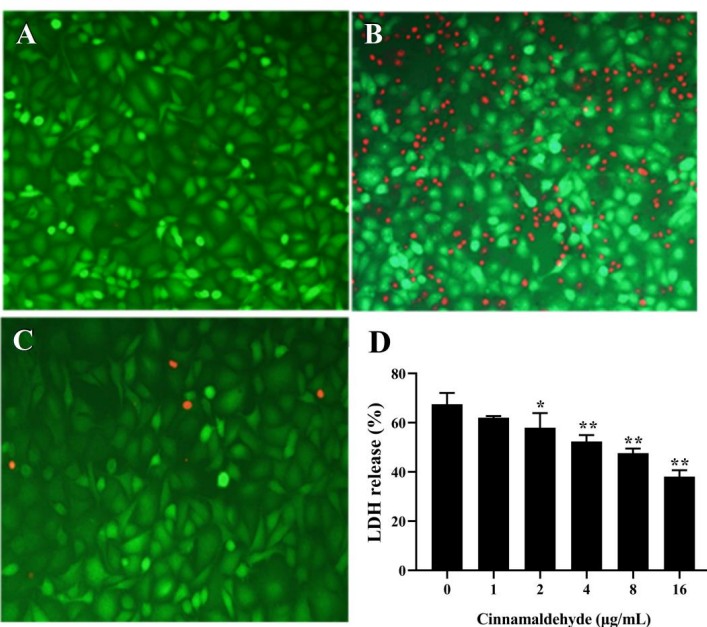

**Figure 7.** The protective effect of cinnamaldehyde on aerolysin-mediated cell injury. (**A**) Untreated cells; (**B**) cells treated with drug-free supernatant; (**C**) cells treated with 16 µg/mL cinnamaldehyde-treated bacterial supernatant; (**D**) LDH release of A549 cells co-cultured with cinnamaldehyde-treated bacterial supernatants; LDH assay was performed in triplicate; data were mean value ± SD. *, $0.01 < p < 0.05$ and **, $p < 0.01$.

### 3.10. Protective Effect of Cinnamaldehyde on Crucian Carp Infected by A. hydrophila

The results above showed that cinnamaldehyde could significantly suppress the product of virulence factors and biofilm formation and provided protection to A549 cells against cell injury. The findings above revealed that cinnamaldehyde might have therapeutic effects on a fish model challenged with *A. hydrophila*, Therefore, the infection model of crucian carp was established. In the positive control group, swelling showed around the fins and ascites in the abdominal cavity. As shown in Figure 8, deaths occurred on the first day in the cinnamaldehyde-treated group and the positive control group. The mortality of fish in the positive control group was 100% in 7 days (Figure 8), and 45% of the cinnamaldehyde-treated group. The results showed statistical significance by log-rank test analysis. All fish were alive in the negative control group during the course.

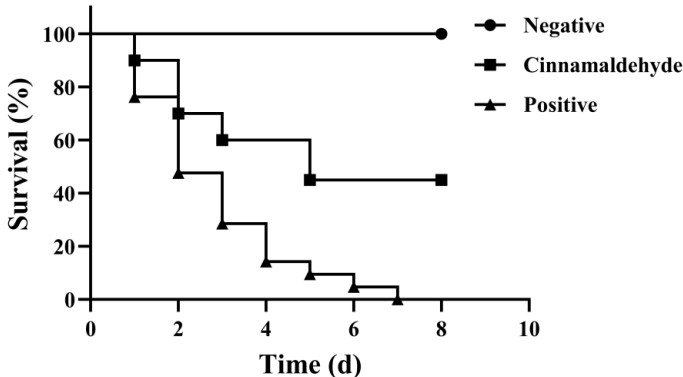

**Figure 8.** Cinnamaldehyde can increase the survival rate of crucian carp infected with *A. hydrophila*. Infected crucian carp were administered with 25 mg/kg of cinnamaldehyde or sterile PBS every 12 h for 3 days; deaths were recorded for 8 days. Treatment with cinnamaldehyde had a significant protective effect on crucian carp infected with *A. hydrophila* when analyzed by log-rank test ($p < 0.0001$).

## 4. Discussion

*A. hydrophila* threatens the healthy development of freshwater aquaculture [17]. Antibiotics are the primary measures for dealing with bacterial diseases in aquaculture, but the occurrence of resistance restricted the practice of antibiotics [18]. Therefore, new therapeutic drugs are needed for increasing bacterial diseases in aquaculture. Studies have demonstrated that herbal medicines have benefits in controlling bacterial infections [19]. A number of studies reported that cinnamaldehyde inhibited the growth of both Gram-positive and Gram-negative bacteria [20,21]. Sapna Rani et al. studied the in vitro antimicrobial activity of cinnamaldehyde using the agar disk diffusion method and showed that cinnamaldehyde could inhibit *Streptococcus agalactiae* at a concentration of 0.66 µg/mL [22]. Xing et al. found that natural cinnamaldehyde could be used as an alternative for fungicide in the field with a MIC of 50 mL/L for *Fusarium verticillioides* [23]. Cinnamaldehyde has been reported to inhibit the growth of bacteria from aquatic sources. Yin et al. found that the MIC of cinnamaldehyde was effective against the growth of *A. hydrophila* CW at 256 µg/mL by disrupting the cell membrane and impacting protein metabolism [24]. Barbara Rossi tested the inhibitory effect of cinnamaldehyde against two *Vibrio* spp.; the results showed that the MICs of cinnamaldehyde were effective against *Vibrio harveyi* and *Vibrio anguillarum* at 1.88 mM and 3.75 mM, respectively [25]. In our study, the MIC of cinnamaldehyde against *A. hydrophila* XS-91-4-1 was 128 µg/mL, which was about two times lower than that reported by Yin et al., and may be due to differences in strains [24].

A variety of naturally extracted compounds have anti-virulence effects on *A. hydrophila*. Jing et al. found that 8 µg/mL of genistein could significantly inhibit the hemolytic activity and biofilm formation of *A. hydrophila* [26]. Sun et al. showed that 25 µg/mL of esculetin could significantly reduce the biofilm formation of *A. hydrophila*, while in our study, 1 µg/mL of cinnamaldehyde could significantly reduce the biofilm formation of *A. hydrophila* [27]. Moreover, cinnamaldehyde has been reported as an inhibitor targeting bacterial virulence and this results in the reduction of the pathogenicity of a number of bacterial pathogens. Ferro et al. reported that cinnamaldehyde could diminish the hemolytic activity and adhere to the latex of *Staphylococcus aureus* (*S. aureus*) at sub-inhibitory concentrations of 0.125 mg/mL, which protected *Galleria mellonella* larvae from *S. aureus* infection [28]. Studies found that cinnamaldehyde could significantly inhibit the biofilms of *Porphyromonas gingivalis* and *Streptococcus mutans* at sub-MIC concentrations [29,30]. Li et al. demonstrated that cinnamaldehyde could interact with the LuxR-type protein of *Pseudomonas fluorescens* and result in the reduction of virulence factors [31]. Mary et al. found that uropathogenic *Escherichia coli* treated with 750 µM cinnamaldehyde could significantly downregulate the expression of the related virulence genes at subinhibitory concentrations, which could decrease the attachment and invasion of bacteria to urinary tract epithelial cells [32]. Moreover, cinnamaldehyde had the same inhibitory effect on the production of virulence factors and the expression of the virulence gene (*aerA*) of *A. hydrophila* in our study.

Gilles Brackman determined the ability of cinnamaldehyde to inhibit AI-2-based QS system in *Vibrio harveyi*; the results showed that cinnamaldehyde and its derivatives could interfere with AI-2 QS at concentrations without anti-bacterial growth and lead to a neutralizing virulence in Artemia shrimp [33]. However, the effect of cinnamaldehyde against the *A. hydrophila* QS system was not reported. Dong et al. found that *A. hydrophila* co-incubated with thymol could significantly downregulate QS-related genes *ahyI* and *ahyR*, and provide protection to channel catfish challenged with *A. hydrophila* [34]. The same methods were used by us and showed that cinnamaldehyde could significantly downregulate the QS-related genes *ahyI* and *ahyR* at the concentrations in our study. Moreover, QS mediated by AHLs regulates the pathogenicity of *A. hydrophila* strains [35]. *C. violaceum* CV026 is a bioreporter that can produce purple violacein in response to AHLs [36]. In our present study, cinnamaldehyde could reduce AHLs production of *A. hydrophila* by using the *C. violaceum* CV026. The results indicated that cinnamaldehyde

could inhibit QS by inhibiting the expression of QS-related genes and the production of AHLs.

In aquaculture, the cinnamon essential oil has been shown to have antibacterial, anesthetic, growth-promoting, and antioxidant effects, indicating a promising use in aquaculture [37]. Moreover, Abdelhamed et al. found that feeding commercial feed mixed with 15 and 20 mg/kg of cinnamaldehyde could increase the survival rate of catfish infected with *Edwardsiella ictaluri* [38]. Faikoh et al. examined zebrafish infected with *Streptococcus agalactiae* and *A. hydrophila*. The zebrafish were immersed in water with 75 μL/L of liposome-encapsulated cinnamaldehyde, which significantly increased the survival rate of the zebrafish (31.1 ± 10.18% and 35.6 ± 3.85%) compared with the control group [39]. Our study showed a higher survival rate of 45% to crucian carp infected with *A. hydrophila* compared with the control group, which may be due to the method of administration and the species of fish. Taken together, our findings provide a candidate alternative drug for *A. hydrophila* infection.

## 5. Conclusions

Cinnamaldehyde could reduce the production of aerolysin, lipase, protease, and AHLs, and inhibit swarming motility and biofilm formation by interfering with the QS of *A. hydrophila* at sub-inhibitory concentrations. qPCR analysis indicated that cinnamaldehyde could downregulate the transcription of aerolysin coding genes (*aerA*) and QS-related genes (*ahyR*, *ahyI*). Moreover, significant therapeutic effects were shown in vitro and in vivo therapeutic assays. Taken together, cinnamaldehyde is a potential QS inhibitor against *A. hydrophila* infection.

**Author Contributions:** J.D. and X.A. designed the experiments. J.D., S.L. and S.Z. carried out the experiments, collected the data, and wrote this manuscript. Y.L., Y.Y. and N.X. checked and revised the manuscript. Q.Y., S.L. and S.Z. contributed data analysis. All authors have read and agreed to the published version of the manuscript.

**Funding:** This work is supported by the National Key R&D Program of China (No. 2019YFD0900104) and the China Agriculture Research System of MOF and MARA (CARS-46).

**Institutional Review Board Statement:** Animal studies were performed under the guidance of the Animal Welfare and Research Ethics Committee at the Yangtze River Fisheries Research Institute (Permission No. YFI-2022DJ-011, 10 June 2022).

**Data Availability Statement:** The data that support the findings of this study are available from the corresponding author upon request.

**Conflicts of Interest:** The authors declare no conflict of interest.

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
