# Peer review of "Cinnamaldehyde Decreases the Pathogenesis of Aeromonas hydrophila by Inhibiting Quorum Sensing and Biofilm Formation"

_fishes, doi:10.3390/fishes8030122_

Round 1

Reviewer 1 Report

This study used several techniques to demonstrate the effect of cinnamaldehyde on Aeromonas hydrophila infection. The results demonstrated clearly the effficacy of this compound to reduce infection by this bacterium, but I am giving some suggestions to improve the manuscript.

The article "The Use of Cinnamon Essential Oils in Aquaculture: Antibacterial, Anesthetic, Growth-Promoting, and Antioxidant Effects". Fishes 2022, 7, 133. https://doi.org/10.3390/fishes7030133

can help improving Introduction and discussion.  

lines 14-15: I suggest ... QS of A. hydrophila  aiming to reduce its pathogenicity.  

lines 25-26: I suggest repplacing the keywords that are in the title

fig. 1 is out of place, it should be addd after its mention in the text. 

line 65: The hemolysis assay is in fig. 1C, not 1B.

table 2: authors must provide efficiency and r2 of the primers as well as accession number of the genes in the GenBank

item 2.13: more details are necessary: volume of the tanks, water quality parameters (oxygen levels, temperature, pH, etc), number of replicates.

The groups were cinnamaldehyde + bacteria, Tween 80 + bacteria, Tween 80? No group with only  cinnamaldehyde ?

line 209: not correct. According to fig. 2A, from 4 to 16 uL/mL, and sentence in lines 211-212 is not necessary, just adjust the sentence in line 209.

lines 222-223: this sentence is not necessary, you have already explained this in the previous sentences.

lines 317-319: add more details. Experiment in vitro?

lines 322-328: authors could hypothesize why there was this difference in MIC for A. hydrophila - different strains?

Discussion from 329 and forward needs some adjustments in the text. All information is there, but I feel that it could be more integrated. Authors are describing different studies one by one, but they could integrate more these results to facilitate to relate them to their work

lines 330-331: delete "Thiago A." and add the concentrations that inhibited the hemolitic activity and protected the larvae.

lines 337-338: this sentence is not necessary, delete it.

line 350: a new paragraph could start with "C. violaceum"

lines 351-354: these sentences are not necessary for discussion and could be removed

References: check format. In some references the titles have all capital letters

Reviewer 2 Report

This is an interesting work. The study is focused on the ability of Cinnamaldehyde to inhibit the virulence factors of the bacteria Aeromonas hydrophila without inhibiting their growth. It is performed at different levels: aerolysin, lipase, protease, swarming motility, biofilm formation, Acyl-homoserine Lactones (AHLs) and QS-related genes. The manuscript is clear and the graphs are of good quality.

In my opinion, the discussion should be developed a bit more, it only describes the results and lists results from other works but does not go into depth. All results should be discussed one by one and the mechanisms that have produced them should be discussed and compared with other articles as appropriate.

Specific comments.

Figure 1- Growth curves don’t see very well; can you try to improve the visualization?

Line 100- please, include a reference.

Line 181- Please explain the program used, if applies. Can you explain better the statistical analyses used (normality…. etc).

Lines 311-327- how can Cinnamaldehyde inhibit biofilm formation, which mechanisms the article proposes? Compared to the doses of other compounds mentioned in the paper, are they higher? lower? the difference in units in the paper makes these details difficult.

Lines 337- Which virulence factors are the same as those used in this study? please compare.
